# The Impact of Post-Analytical Tools on New York Screening for Krabbe Disease and Pompe Disease

**DOI:** 10.3390/ijns6030065

**Published:** 2020-08-14

**Authors:** Monica M. Martin, Ryan Wilson, Michele Caggana, Joseph J. Orsini

**Affiliations:** Wadsworth Center, New York State Department of Health, Newborn Screening Program, David Axelrod Institute, 120 New Scotland Ave., Albany, NY 12201, USA; monica.martin@health.ny.gov (M.M.M.); ryan.wilson@health.ny.gov (R.W.); michele.caggana@health.ny.gov (M.C.)

**Keywords:** newborn screening, Krabbe, Pompe, post-analytical tools, Collaborative Laboratory Integrated Reports

## Abstract

New York uses a two-tier assay to screen newborns for Krabbe disease and Pompe disease. Individual enzyme activities are measured in the first-tier, and specimens from newborns with low activity are reflexed to second tier Sanger sequencing of the associated gene. Using only this two-tiered approach, the screen positive and false positive rates were high. In this study, we added an additional step that examines the activity of four additional lysosomal enzymes. Results for all enzymes are integrated using the multivariate pattern recognition software called Collaborative Laboratory Integrated Reports (CLIR) to assess the risk for disease. Results after one year of screening using the new algorithm are compared to the prior year of screening without consideration of the additional enzymes and use of CLIR. With CLIR the number of babies referred for Krabbe disease was reduced by almost 80% (from 48 to 10) and the number of babies referred for Pompe disease was reduced by almost 32% (22 to 15).

## 1. Introduction

New York screens all newborns for Krabbe and Pompe diseases; both are autosomal recessive disorders. Krabbe has a predominate early infantile form affecting 85–90% of diagnosed individuals. Whereas Pompe disease is more clinically heterogeneous with a higher percentage of cases being detected later in life. In NY, screening for Krabbe and Pompe diseases was originally accomplished using a two-tier strategy of enzyme activity followed by molecular analysis (Sanger sequencing of the *GALC* and *GAA* genes, respectively). Both diseases are autosomal recessive and variants must be *in trans* for symptoms to manifest. Reduced enzyme activity (GALC for Krabbe and GAA for Pompe) is present in patients diagnosed with the disease, but activity ascertained from a dried blood spot alone is not a distinguishing characteristic and leads to a high rate of false positives. There are multiple potential reasons for this, examples include enzyme lowering benign gene variants, age of the newborn at the time of specimen collection (white cell levels and hematocrits vary in the newborn period [1]), inhibitors in the blood [2], and specimen handling and transport issues. All of these factors can contribute to false positive screens.

New York initially incorporated second-tier molecular testing to reduce the number of false positive screens for Krabbe and Pompe diseases, as at the time it was the only available approach to increase the specificity of the screen. In the NY two-tier algorithm, newborn dried blood spot samples with an enzyme activity less than the cutoff were Sanger sequenced. Only those newborns with one or more disease-causing variants or variants of unknown significance (VOUS) were referred for a follow-up diagnostic evaluation. The false positive rate was reduced because newborns carrying only non-disease-causing benign variants were not referred. With this approach the number of Krabbe and Pompe referrals was reduced by 46% [3] and 22%, respectively (unpublished data). The conservative approach of reporting newborns with only one potentially disease-causing variant was chosen since there is a chance that a second variant could be missed in Sanger sequencing. Examples include deletions/duplications, cryptic splice sites, and allele dropout. Consequently, after diagnostic enzyme testing, many of these screen positive infants were carriers only. In most of these cases, a pathogenic variant and one or more pseudodeficiency variants were detected. The combination of these molecular changes and presumed other inherent variables present in the dried blood spot resulted in enzyme activities below the cutoff yielding a false positive result.

In 2015 the Commonwealth of Kentucky passed a bill mandating screening for Krabbe disease. To accelerate implementation, the Kentucky Department of Public Health contacted the Biochemical Genetics Laboratory at Mayo Clinic (Rochester, Minnesota) to outsource the screening. The Mayo staff were concerned that screening for Krabbe disease using the conventional method of testing for a single marker (GALC activity) would lead to too many false positive results. Instead, Mayo made a counterproposal to obtain a profile of six lysosomal enzyme activities per newborn inclusive of the enzymes for Pompe disease and mucopolysaccharidosis type I (MPS I). Their approach relies on six covariate-adjusted enzyme activities integrated by all informative permutations of calculated ratios among them [4]. If after measuring the six enzyme activities a profile suggestive of the targeted condition is indicated, the specimen was again tested for all six enzyme activities plus four C20:0–C26:0 lysophosphatidylcholines, which are used to identify newborns at risk for adrenoleukodystrophy. The data from the 10-plex assay are used to further evaluate the result. By measuring all of these markers, the Mayo Clinic could get the most power out of their in-house developed CLIR software. CLIR evaluates each marker and its ratio to all the other measured informative markers for specific birthweight and age at collection reference ranges to help reduce the false number of positive screens [4]. The results using CLIR were notable, with no false positives reported for Krabbe or Pompe diseases after testing 55,161 specimens. This prompted NY to implement a similar approach using CLIR in screening for Krabbe and Pompe diseases. Herein, we report how the use of CLIR tools affected screening for these two diseases in New York.

## 2. Materials and Methods

Two types of New York-specific tools for the assessment of Krabbe and Pompe diseases were established based on tools already available in CLIR. New York-specific tools are required because the existing tools in use by Mayo include some LSD enzyme and LPC markers which are not measured in New York newborn screening. Since GALC and GAA activities and C26:0-LPC concentration are measured on all New York newborns, and are included in the Mayo tools, we chose to use this subset of markers in the first set of New York-specific tools (“NY3-plex”). We also created 7-plex tools, which include the activities of four additional enzymes for Gaucher disease, Niemann Pick A/B disease, mucopolysaccharidosis type I, and Fabry disease (enzymes abbreviated: ABG, ASM, IDUA, and GLA, respectively). These additional enzymes are used for calculation of ratios only and are not otherwise evaluated. Table 1 shows the markers and ratios that are used in these two tools. Activities of these four additional enzymes are measured on specimens with initial GALC and/or GAA enzyme activity below their cutoffs (see below). The extra enzyme analyses were chosen for testing in order to further improve the assessment of the NY3-plex positive specimens.

Enzyme results and required covariates (age at time of collection, sex, and birthweight) for true negative cases (negative for all disorders on the NY screening panel) were compiled in addition to both true positive and false positive NY cases for Krabbe and Pompe diseases. Several of the cases positive for Krabbe disease in our system were missing marker values because they were tested prior to the implementation of X-Adrenoleukodystrophy (ALD) and/or Pompe disease screening. Due to the limited number of these positive specimens, those missing the additional markers were pulled from the archive and analyzed to obtain the missing data required for the 7-plex tool assessment. A potential weakness of this study is that archived samples may have reduced enzyme activities and/or C26:0-LPC marker concentrations due to the age of the specimen at the retrospective analysis.

After the supplementary testing was completed, 328,604 true negative cases were uploaded to CLIR with all required covariates to establish the reference range for GAA, GALC, and C26:0-LPC. Additionally, a total of 22 Krabbe and 18 Pompe true positive cases were uploaded with all marker concentrations. Cases were classified as true positive if they had two known likely pathogenic variants, or one known likely pathogenic variant, a variant of unknown significance, and an abnormal diagnostic enzyme result in leukocytes. These cases included newborns with the classic form of each disease as well as newborns that are thought to be at most risk for the disease in childhood; the latter group was defined as those newborns with a variant observed in early infantile disease (generally a nonsense variant) and a variant observed in patients with late onset disease. A total of 13 Krabbe and 11 Pompe cases were uploaded for the pool of false positive cases. Cases were classified as false-positive if they had no known likely pathogenic variants, or two variants associated with late-onset disease and a normal diagnostic enzyme result in leukocytes.

The enzyme activities were measured using a modified version of the method reported by Elliot et al. [5] and C26:0-LPC concentrations were measured for each newborn using a modified version of the method of Tortorelli et al. [6]. After completion of testing, the results for GAA activity, GALC activity, and C26:0-LPC concentration were uploaded into the Program’s Laboratory Information Management System (LIMS). Subsequently, the analyte activities/concentrations and demographic information including date/time of birth, date/time of specimen collection, gestational age, sex and birthweight for each newborn’s specimen were exported from the LIMS into an Excel data template for CLIR compatible covariate creation. The resulting properly formatted CSV file was uploaded and assessed using the NY-specific CLIR 3-plex tool runner (TR) applications for Krabbe and Pompe diseases. Note that the dates/times of birth and sample collection are used to calculate the age of the infant at the time of specimen collection, however, are not included in the final CSV file to ensure privacy.

TR calculates a score for each sample based on the amount of marker/ratio overlap with reference ranges. Points are scored, in increasing type, as the gap between marker/ratio and reference range increases, with zero scores being reserved for samples with all marker/ratios within reference ranges. Specimens with a 3-plex TR score greater than zero for either disorder were further assessed using the 3-plex dual scatter plot (DSP) specific for the disorder. The DSP assigns each specimen to one of three categories; false positive, indeterminate, or disease positive. Note that, any specimen, per the NY algorithm, with % daily mean activity (DMA) lower than the retest cutoff (≤16% GALC or ≤20% GAA) and greater than a screen positive first-tier cutoff (<12% GALC or <15% GAA) that are dismissed by CLIR (TR score of zero or TR score greater than zero and DSP assessment of false positive) were considered as screen negative for the corresponding disorder.

Specimens below the first-tier cutoff for either disorder, with no prior normal screening result on record, were re-tested in duplicate using a multiplex, FIA-MS/MS, 6-plex enzyme assay which measures the activities of GLA, IDUA, ABG, and ASM in addition to GALC and GAA. Following the re-test, the 6-plex enzyme activities were combined with C26:0-LPC concentrations and demographic covariates for CLIR assessment using the NY-specific 7-plex TR tools. Specimens with a 7-plex TR score greater than zero for either disorder were further assessed using the 7-plex DSP specific for the disorder. Those with a TR score of zero or a DSP assessment of false positive were considered screen negative for the corresponding disorder while specimens with a DSP assessment of indeterminate or positive were sent for molecular analysis. Following molecular analysis and interpretation, specimens with one known or likely pathogenic variant or a single variant of unknown significance were considered screen positive and the newborn was referred to a Specialty Care Center while all others were considered screen negative.

CLIR tools were used as described above during one year of live screening. This year of data was then compared to the one-year period immediately prior where only cutoffs were used for the risk assessment.

## 3. Results

A summary comparing the results from screening with and without the use of CLIR over the two-year period is shown in Table 2. The total number (#) of specimens assessed in each period were 260,620 and 262,467 specimens, respectively. The use of CLIR 3-plex tools decreased the number of re-tests after initial screening (1163 to 237 for GALC and 346 to 161 for GAA). Similarly, the number of specimens sent for molecular analysis using NY% DMA cutoffs alone versus CLIR 7-plex tools decreased from 90 to 13 for GALC. The decrease for GAA was more modest (24 to 20). Finally, the number of babies referred for Krabbe disease was reduced by almost 80% (from 48 to 10) with the use of combined CLIR tools. The number of babies referred for Pompe disease was reduced by almost 32% (22 to 15). Perhaps circumstantially, the number of possible disease cases, based on diagnostic testing and clinical evaluation were similar across both one-year time periods, loosely indicating that there were no missed cases during screening with CLIR. The use of NY cutoff values alone yielded six Krabbe disease cases and twelve Pompe disease cases while incorporation of the CLIR assessment tools resulted in five and thirteen cases, respectively.

## 4. Discussion

Since screening began in NY (2005 for Krabbe and 2014 for Pompe) through 2016, there have been 462 Krabbe referrals (~2.5 million newborns screened) and 89 Pompe referrals (~526 K newborns screened). These high referral rates 0.017% and 0.015% led to a desire to improve screening and decrease the adverse impact of an unnecessary screen positive result on families. For most of this screening period, only the percent of daily mean cutoff was used for each individual enzyme (12% and 15% for Krabbe and Pompe diseases, respectively) to determine which samples required re-testing and/or second-tier Sanger sequencing. Although Sanger sequencing reduced the number of newborns referred, many were still unnecessarily referred. Most of these newborns had one reportable variant and were likely carriers with a pseudodeficiency variant(s) *in trans.* Specimens from these infants flagged originally due to low enzyme activity on the initial screening test.

Of all the possible variables influencing the measured DBS enzyme activities, birthweight and age of the newborn at collection are known and can be used in establishing covariate reference ranges in CLIR. To illustrate the birthweight and age dependency, Table 3 shows the mean enzyme activities and percent of mean values for 32,092 samples (collected as part of a pilot screening in NY [7]) stratified by age at collection and birthweight. The table also shows the average activity and average % of mean activity values for each enzyme for the subset of samples where IDUA is <20%, GALC is <20%, and GALC is >300%. Of the six enzymes, GALC activities vary the most with birthweight. The average GALC activity of newborns with birthweights less than 1500 g is 13.9 µmol/L/h vs. 5.9 µmol/L/h for all samples tested. For samples collected from normal birthweight infants (>2500 g) from babies >2 weeks of age, the mean GALC activity is 3.6, or 61.0% of the GALC mean of all samples tested (5.9 µmol/L/h). The observed variability in mean GALC activity and percent daily mean as a function of birthweight and age illustrates how the use of a single cutoff for GALC could lead to a high rate of false positive results. Notably, the GALC activity is significantly reduced when the specimen is collected from an older infant (>2 weeks). In addition, we often observe low activities across the panel when a single enzyme flags for low activity. See examples below where IDUA and GALC are <20% and the others are all less than 100%, e.g., for the pool of samples with <20% IDUA activity, the average GALC activity is 67.4% compared to the GALC mean activity for the entire sample set (4.0 vs. 5.9).

Since screening began in NY, several improvements have been made to the assay that made it easier to implement multiplexed enzyme testing. For the first 10 years of screening for Krabbe disease, only the measured percent of daily mean GALC activity was used to move a specimen to a second-tier DNA sequence analysis. Once the laboratory acquired the ability to multiplex six LSD enzymes, the sample “quality” could be assessed as well. For unaffected newborns, in general, when one enzyme activity is low all the other enzymes tend to have low normal activities as well (presumably from the aforementioned combination of varied leukocyte counts, varied hematocrits, or sample quality issues). Whereas, for specimens from newborns diagnosed with an LSD, the relative activity of the screen positive enzyme will be much lower than all the others.

When one enzyme activity result is below the set single enzyme cutoff (near each of their single enzyme cutoffs) and the other five enzymes are also low (presumably from a combination of varied leukocyte counts, varied hematocrits, or sample quality issues); it is unlikely that the newborn has elevated risk for an LSD. In these cases, some laboratories request a repeat specimen, which requires a new sample to be taken, shipped, and tested. In our experience, the acquired repeat sample often produces a similar result, with multiple low measured activities, and just delays the final assessment. Risk assessment becomes more challenging when a sample has one enzyme activity result below the set single enzyme cutoff and the others are moderately decreased and has a non-standard age at the time of collection. Minter-Baerg et al. showed that by using reference ranges established with these variables, screening for Krabbe and Pompe diseases could be done with no false positives. [4] While second-tier biochemical testing was also used in this work, their report demonstrated the utility of the CLIR tools in the screening algorithm for Krabbe and Pompe diseases. Data from a one-year period in New York demonstrated that the CLIR tools can reduce the number of babies requiring second-tier DNA sequencing, with a presumed equivalent identification of true positive cases. More importantly, the CLIR tools greatly reduced the number of newborns referred, with Krabbe disease referrals decreased from 0.018% to 0.0038% and Pompe disease referrals decreased from 0.0084% to 0.0057%. This reduced work for follow-up and the associated anxiety to families; thus, allowed stretched programs and clinicians to focus efforts on babies that are more likely affected with these diseases.

## Figures and Tables

**Table 1 IJNS-06-00065-t001:** Markers and ratios used in NY3-plex and NY7-plex.

CLIR Tool	Marker	Ratio 1	Ratio 2	Ratio 3	Ratio 4	Ratio 5	Ratio 6
Krabbe-3plex	GALC	C26/GALC	GALC/GAA	n/a	n/a	n/a	n/a
Pompe-3plex	GAA	C26/GAA	GALC/GAA	n/a	n/a	n/a	n/a
Krabbe-7plex	GALC	C26/GALC	IDUA/GALC	GALC/ASM	GALC/GAA	GALC/ABG	GALC/GLA
Pompe-7plex	GAA	C26/GAA	ABG/GAA	IDUA/GAA	GALC/GAA	GAA/ASM	GAA/GLA

**Table 2 IJNS-06-00065-t002:** Number of specimens at each stage of screening during the first year of Collaborative Laboratory Integrated Reports (CLIR) tool use compared to the year prior. Disease refers to the number of true cases.

Disorder	Total Assessed		# Re-Tested		# Sequenced		# Referred		Disease	
	w/Cutoffs	w/CLIR	w/Cutoffs	w/CLIR	w/Cutoffs	w/CLIR	w/Cutoffs	w/CLIR	w/Cutoffs	w/CLIR
Krabbe	260,620	262,467	1163	237	90	13	48	10	6	5
Pompe	260,620	262,467	346	161	24	20	22	15	12	13

# is number.

**Table 3 IJNS-06-00065-t003:** Average enzyme activities (µmol/L/h) and percent of mean values (listed as %GALC, %GAA, etc.) of all samples tested (32,092) and stratified as indicated in the first column. The IDUA and GALC <20% and GALC >300% rows show the average activities and % of mean activities for each enzyme listed. Other enzymes are included for comparison of trends.

Population\Subset Population of Samples	GALC	GAA	IDUA	GLA	GBA	ASM	%GALC	%GAA	%IDUA	%GLA	%GBA	%ASM
32,092 samples tested	5.9	9.8	7.0	13.3	13.1	7.2	100.0	100.0	100.0	100.0	100.0	100.0
<1500 g at birth	13.9	11.5	7.1	26.8	12.1	6.2	235.3	117.8	102.5	201.4	92.5	86.1
>2 weeks of age	5.3	9.1	7.1	12.3	8.3	7.8	89.5	93.6	101.8	92.2	63.3	108.1
>2 weeks and >2500 g	3.6	7.7	7.1	9.1	7.5	9.3	61.0	79.4	101.5	68.3	57.2	129.2
Samples with IDUA <20%	4.0	4.6	0.9	8.4	5.5	5.4	67.4	46.9	13.0	63.3	42.2	75.5
Samples with GALC <20%	1.0	6.8	5.4	7.2	8.8	7.1	16.1	69.8	76.9	54.1	66.9	98.5
Samples with GALC >300%	27.4	12.6	8.1	41.1	17.8	6.2	463.6	129.6	116.1	308.6	135.9	86.1

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
