# Peer review of "The Impact of Post-Analytical Tools on New York Screening for Krabbe Disease and Pompe Disease"

_2409-515X, 2020, doi:10.3390/ijns6030065_

Round 1

Reviewer 1 Report

Title:  Include “Post-analytical tools” in the title for clarity to those who may be unfamiliar with CLIR

No email address provided for corresponding author

Abstract:

-How many enzymes were included total?  The abstract says 6 additional, but the text of the article has 6 total

-Include a sentence at the end of the abstract summarizing the results of the study, as the abstract doesn’t currently include any results / outcome information.

Introduction:

-Include basic clinical information or references for disorders, particularly to highlight variable onset of symptoms and the difficulties correlating variants with clinical onset - one of the main limitations of using molecular alone as the second tier test

-Line 20:  would “strategy” be a better word than “assay”?

-the phrase “enzyme lowering gene variants” is confusing.  It appears that it is referring to pseudodeficiency, but the phrasing could refer to disease causing variants. Please clarify

-p2 line 52 - “abnormal initial result” is vague - “profile suggestive of the targeted condition”, as decreased enzyme activity is abnormal, but not sufficient cause for referral to second tier testing

-p2 line  56 - please add that the LPCs are intended for the identification of XALD

P2 line 58 - “the results were notable” is vague - please specify high PPV, low FP rate etc

Materials and Methods

-Naming of tools:  please clarify what analytes are included in the 3-plex and 7plex tools (including ratios, possibly included as supplementary table)

Lines 69-70:  “assess all initially screened specimens….”  -> vague, perhaps “interpret first tier testing results” to keep the first / second tier nomenclature consistent throughout

Lines 82 - 84:  references appear to be missing

Overall:  A flow chart would be helpful for following the interpretation steps for the reporting  There seem to be two pathways to the second tier test?

Line 112:  Needs either a basic explanation or an explicit reference to CLIR scores for appropriate context

Line 119:  This is the first mention I could find of a “molecular cutoff” and it isn’t well-explained

Results:

How did the numbers end up at 264,000 for each year?  Was this purposely selected, and why?  If it isn’t all samples screened over that time period, how were they selected?

Line 145:  Please clarify that there were no known “missed” cases during CLIR screening, that the variation observed in the number of cases would not be unexpected as year to year variation.

Table 1:  Consider a more descriptive label for the column than “Before”, perhaps “Cutoff based”?  Also potentially change “disease” to “True Positive”

Table 2:  There is no caption present.  This table might be more clear if it just focused on IDUA, GAA and GALC activities, as they are the ones showing the most pronounced effects, and used for identifying targeted conditions.  Units are also needed for enzyme activity columns.

Line 182:  Consider “for unaffected infants” rather than “newborn screening samples in general”

Lines 193-194:  Add the reference to the Baerg paper.

Lines 197-200:  There should be some mention of the presumed equivalent performance for identification of TP cases (as much as possible given the limited time period of follow-up)

Author Response

Dear Reviewers,

Thank you all for your helpful comments, below are our responses.  Hope these changes are satisfactory.

Sincerely, Joseph Orsini

Reviewer 1 Comments and author responses:

Title:  Include “Post-analytical tools” in the title for clarity to those who may be unfamiliar with CLIR

No email address provided for corresponding author

Thank you for pointing out this over sight, added contact info. and corresponding author.

Abstract:

-How many enzymes were included total?  The abstract says 6 additional, but the text of the article has 6 total

Thanks, corrected to four additional for 6 total.

-Include a sentence at the end of the abstract summarizing the results of the study, as the abstract doesn’t currently include any results / outcome information.

Thank you, added at end: “With CLIR the number of babies referred for Krabbe disease was reduced by almost 80% (from 48 to 10) and the number of babies referred for Pompe disease was reduced by almost 32% (22 to 15).”

Introduction:

-Include basic clinical information or references for disorders, particularly to highlight variable onset of symptoms and the difficulties correlating variants with clinical onset - one of the main limitations of using molecular alone as the second tier test

Added: New York screens all newborns for Krabbe and Pompe diseases, both are autosomal recessive disorders. Krabbe has a predominate early infantile form affecting with 85-90 % of diagnosed individuals. Whereas for Pompe disease is more clinically heterogeneous with a higher percentage of cases being detected later in life. We discuss issue with second tier genotyping later in the manuscript.

-Line 20:  would “strategy” be a better word than “assay”?

Changed to “strategy”.

-the phrase “enzyme lowering gene variants” is confusing.  It appears that it is referring to pseudodeficiency, but the phrasing could refer to disease causing variants. Please clarify

Added “benign” to sentence.

-p2 line 52 - “abnormal initial result” is vague - “profile suggestive of the targeted condition”, as decreased enzyme activity is abnormal, but not sufficient cause for referral to second tier testing

Added: “a profile suggestive of the targeted condition is indicated”

-p2 line  56 - please add that the LPCs are intended for the identification of XALD

Added “which are used to identify newborns at risk for adrenoleukodystrophy”.

P2 line 58 - “the results were notable” is vague - please specify high PPV, low FP rate etc

Added: “with no false positives reported for Krabbe or Pompe diseases after testing 55,161 specimens”.

Materials and Methods

-Naming of tools:  please clarify what analytes are included in the 3-plex and 7plex tools (including ratios, possibly included as supplementary table) 

This is done, however, added table to the document.

Lines 69-70:  “assess all initially screened specimens….”  -> vague, perhaps “interpret first tier testing results” to keep the first / second tier nomenclature consistent throughout

                Changed to: “The purpose of the NY3-plex tools is to interpret first-tier testing results to determine which specimens are positive and require retesting”.

Lines 82 - 84:  references appear to be missing

                We noticed that the sentences with missing references were redundant with the same information in the next paragraph, so the two sentences were removed from the text.

Overall:  A flow chart would be helpful for following the interpretation steps for the reporting  There seem to be two pathways to the second tier test?

                Paragraph was modified per highlighted area (lines 120 – 122) to clarify.  There is only one way to a second-tier test.  Hopefully this makes it unnecessary to include a flow chart.

                Line 112:  Needs either a basic explanation or an explicit reference to CLIR scores for appropriate context

Added: TR calculates a score for each sample based on the amount of marker/ratio overlap with reference ranges.  Points are scored, in increasing type, as the gap between marker/ratio and reference range increases, with zero scores being reserved for samples with all marker/ratios within reference ranges.

Line 119:  This is the first mention I could find of a “molecular cutoff” and it isn’t well-explained

                Renamed “molecular cutoff” as ”first tier positive cutoff” to clarify.

Results:

How did the numbers end up at 264,000 for each year?  Was this purposely selected, and why?  If it isn’t all samples screened over that time period, how were they selected?

                Thank you for catching this error. The actual numbers each year were 260,620 (w/ cutoffs) and 262,467 (w/ CLIR).  Table 1 and the text have been adjusted to reflect the correction.

Line 145:  Please clarify that there were no known “missed” cases during CLIR screening, that the variation observed in the number of cases would not be unexpected as year to year variation.

                We stated that “Importantly” the number of possible disease cases, based on diagnostic testing and clinical evaluation were similar across both 1-year time periods….  Changed wording to “Perhaps circumstantially”, the number of possible cases….

Table 1:  Consider a more descriptive label for the column than “Before”, perhaps “Cutoff based”?  Also potentially change “disease” to “True Positive”

Changed from “before” to “w/cutoffs”

Table 2:  There is no caption present.  This table might be more clear if it just focused on IDUA, GAA and GALC activities, as they are the ones showing the most pronounced effects, and used for identifying targeted conditions.  Units are also needed for enzyme activity columns.

We had included the caption for Table 2 with the original submission, apparently it didn’t make it into the version for review. The caption has been added and it provides the units for enzyme activity. Caption states that the other enzymes are included for comparison of trends.

Line 182:  Consider “for unaffected infants” rather than “newborn screening samples in general”

Made this change.

Lines 193-194:  Add the reference to the Baerg paper.

                Done.

Lines 197-200:  There should be some mention of the presumed equivalent performance for identification of TP cases (as much as possible given the limited time period of follow-up)

                Added to sentence: “with a presumed equivalent identification of true positive cases.”

Reviewer 2 Report

This is an important paper and shows the enormous strength of CLIR when applied to the complex screening environment of Pompe and Krabbe.

The paper is important in showing a major reduction in the work of the lab, but importantly in the reduction of referrals which is a huge problem in the United States with its broad screening programs.

Two corrections need to be made in the MS: there need to be reference numbers associated with attached references supplied on line 84 and 84 which currently only have descriptions of the material to be cited.

Author Response

Dear Reviewers,

Thank you all for your helpful comments, below are our responses.  Hope these changes are satisfactory.

Sincerely, Joseph Orsini

Comments and Suggestions for Authors

This is an important paper and shows the enormous strength of CLIR when applied to the complex screening environment of Pompe and Krabbe.

The paper is important in showing a major reduction in the work of the lab, but importantly in the reduction of referrals which is a huge problem in the United States with its broad screening programs.

Two corrections need to be made in the MS: there need to be reference numbers associated with attached references supplied on line 84 and 84 which currently only have descriptions of the material to be cited.

Response:           Thank you for your nice comments. I Noticed that the sentences with missing references were redundant with information in the next paragraph. These two sentences were removed from the text.

Reviewer 3 Report

1.  I would like to see examples of the 3-plex and 7-plex DSP.

2.  Do the authors have any ideas as to why the decreases in the number of retests and specimens referred to molecular testing were lower for GAA than for GALC?

3.  I suggest mentioning the referral rates for the second year when the CLIR tools were used.

4.  In line 179 the authors state that improvements were made to the assay.  Would it be more accurate to state that the improvements were to the testing algorithm?

5.  It was touched on in the Introduction but I think it should be mentioned in the Discussion the reasons that specimens with low activity of one enzyme frequently have low activities of multiple enzymes.

6.  Since this study was prompted by the experience in Kentucky, it would be nice to see a data comparison of referral rates with Kentucky and any other newborn screening programs that have published data on the use of CLIR tools either prospectively or retrospectively.

Author Response

Dear Reviewers,

Thank you all for your helpful comments, below are our responses.  Hope these changes are satisfactory.

Sincerely, Joseph Orsini

Reviewer 3 comments and author responses:

Comments and Suggestions for Authors

  1. I would like to see examples of the 3-plex and 7-plex DSP.

We attempted to provide examples, but we no longer use the 3-plex tool and it is no longer available in the CLIR tools (we have been screening for MPS I and added IDUA and made a 4-plex tool that is used instead. Hopefully, for interested readers, the example of DSPs in the Kentucky paper will suffice.

  1. Do the authors have any ideas as to why the decreases in the number of retests and specimens referred to molecular testing were lower for GAA than for GALC?

                We are not sure, certainly GALC activity has larger age and birthweight dependency than GAA and CLIR does a good job or taking these variables into account (age/weight specific reference ranges). We do notice a fair number of samples that we are able to exclude from second tier for GAA based on previous samples analysis, or if baby is in NICU and we have a second sample in hand where one or the other of the two specimens has normal activity.

  1. I suggest mentioning the referral rates for the second year when the CLIR tools were used.

                Added to last paragraph: “with Krabbe disease referrals decreased from 0.018% to 0.0038% and Pompe disease referrals decreased from 0.0084% to 0.0057%.”

  1. In line 179 the authors state that improvements were made to the assay. Would it be more accurate to state that the improvements were to the testing algorithm?

                To clarify that it was improvements to the assay by adding to end of sentence: “that made it easier to implement multiplexed enzyme testing”.

  1. It was touched on in the Introduction but I think it should be mentioned in the Discussion the reasons that specimens with low activity of one enzyme frequently have low activities of multiple enzymes.

                To line 203 added: “(presumably from a combination of varied leukocyte counts, varied hematocrits, or sample quality issues);”.

  1. Since this study was prompted by the experience in Kentucky, it would be nice to see a data comparison of referral rates with Kentucky and any other newborn screening programs that have published data on the use of CLIR tools either prospectively or retrospectively.

                This request was not completed since KY referrals also had second tier biochemical and NY only used CLIR. This was stated in the last paragraph of the discussion section.